# Two New Apotirucallane-Type Triterpenoids from the Pericarp of *Toona sinensis* and Their Ability to Reduce Oxidative Stress in Rat Glomerular Mesangial Cells Cultured under High-Glucose Conditions

**DOI:** 10.3390/molecules25040801

**Published:** 2020-02-12

**Authors:** Di Liu, Rong-shen Wang, Lu-lu Xuan, Xiao-hong Wang, Wan-zhong Li

**Affiliations:** School of Pharmacy, Weifang Medical University, Weifang 261053, Shandong Province, China; ld0928928@163.com (D.L.); wangrs199012@126.com (R.-s.W.); xuanll1995@163.com (L.-l.X.)

**Keywords:** *Toona sinensis* (A. Juss.) Roem, apotirucallane-type triterpenoid, cycloartane-type triterpenoid, rat glomerular mesangial cells, oxidative stress

## Abstract

Hyperglycemia is a strong risk factor for chronic complications of diabetes. Hyperglycemic conditions foster not only the production of reactive oxygen species (ROS), but also the consumption of antioxidants, leading to oxidative stress and promoting the occurrence and progression of complications. During our continuous search for antioxidant constituents from the pericarp of *Toona sinensis* (A. Juss.) Roem, we isolated two previously unreported apotirucallane-type triterpenoids, toonasinensin A (**1**) and toonasinensin B (**2**), together with five known apotirucallane-type triterpenoids (**3**–**7**) and two known cycloartane-type triterpenoids (**8**–**9**) from the pericarp. Compounds **8**–**9** were obtained from *T. sinensis* for the first time. Their structures were characterized based on interpretation of spectroscopic data (1D, 2D NMR, high-resolution electrospray ionization mass spectra, HR-ESI-MS) and comparison to previous reports. Compounds (**2**, **4**, **6**, **7**, and **9**) were able to inhibit proliferation against rat glomerular mesangial cells (GMCs) cultured under high-glucose conditions within a concentration of 80 μM. Compounds (**2**, **6**, and **7**) were tested for antioxidant activity attributable to superoxide dismutase (SOD), malondialdehyde (MDA), and ROS in vitro, and the results showed that compounds (**2**, **6**, and **7**) could significantly increase the levels of SOD and reduce the levels of MDA and ROS. The current studies showed that apotirucallane-type triterpenoids (**2**, **6**, and **7**) might have the antioxidant effects against diabetic nephropathy.

## 1. Introduction

Diabetic nephropathy (DN) is one of the most common complications of diabetes, and the occurrence and development of oxidative stress play an important role [1,2]. High-glucose levels can increase the accumulation of reactive oxygen species (ROS) by promoting the generation of ROS and inhibiting the activity of antioxidant enzymes in cells. Excessive oxidative stress could lead to inflammation, fibrosis, the cell apoptosis, and cell damage and death, which are also considered as an important pathological change in DN [3]. There is an urgent need to identify previously unreported antioxidant constituents.

*Toona sinensis* (A. Juss.) Roem is traditional Chinese medicine belonging to the genus *Toona* and the family Meliaceae, which is widely distributed across Asia [4,5]. *T. sinensis* is rich in triterpenoids, phenols, alkaloids, saponins, sterols, coumarin, and anthraquinone [6,7,8,9]. *T. sinensis* had been shown to possess a variety of pharmacological activities, including antioxidant [10], anti-inflammatory [11], bactericidal [12], analgesic [13], antiviral [14], and regulating blood sugar level [14,15].

No study has yet reported the extraction and separation of chemical components in the pericarp of *T. sinensis*, which is often discarded as waste. In the search for potential antioxidant constituents, we performed a chemical investigation and bioactive evaluation of compounds from the pericarp of *T. sinensis*. We described the isolation and structures of two previously unreported apotirucallane-type triterpenoids, toonasinensin A (**1**) and toonasinensin B (**2**), and those of five known apotirucallane-type triterpenoids (**3**–**7**) and two known cycloartane-type triterpenoids (**8**–**9**) (Figure 1). The reducing oxidative stress activities of compounds **1**–**9** were evaluated in rat glomerular mesangial cells (GMCs) cultured under high-glucose conditions.

## 2. Results and Discussion

Compound **1** was obtained as a white amorphous powder. Its molecular formula was assigned as C_37_H_62_O_7_ by high-resolution electrospray ionization mass spectra (HR-ESI-MS) (Appendix A) at *m*/*z* 617.44177 [M–H]^−^ (calculated. 617.44228). The ^1^H-NMR spectrum (Appendix A) exhibited an olefinic proton at *δ*_H_ 5.44 (1H, broad singlet (br s), H-15); five hydroxy methine protons at *δ*_H_ 4.92 (1H, d, *J* = 3.3 Hz, H-21), 4.65 (1H, m, H-3), 4.27 (1H, m, H-23), 3.94 (1H, br s, H-7), and 3.24 (1H, m, H-24); one hydroxy methylene group at *δ*_H_ 3.76 (1H, m, 21-O*CH*_2_CH_3_a), 3.46 (1H, m, 21-O*CH*_2_CH_3_b); and 10 methyl groups at *δ*_H_ 1.26 (3H, s, H-26), 1.20 (6H, s, H-27, 21-OCH_2_*CH*_3_), 1.11 (3H, s, H-18), 1.09 (3H, s, H-29), 0.98 (3H, s, H-19), 0.96 (6H, s, H-4′, 5′), 0.93 (3H, s, H-30), and 0.86 (3H, s, H-28). The ^13^C-NMR spectrum (Appendix A) displayed one carbonyl carbon at *δ*_C_ 174.6 (C-1′); one double bond group at *δ*_C_ 162.7 (C-14), 120.3 (C-15); five hydroxy methine signals at *δ*_C_ 109.7 (C-21), 79.7 (C-3), 78.4 (C-24), 77.1 (C-23), and 73.7 (C-7); one hydroxy methylene signal at *δ*_C_ 64.8 (21-O*CH*_2_CH_3_); and 10 methyls at *δ*_C_ 28.6 (C-29), 27.5 (C-28), 27.0 (C-26), 25.3 (C-27), 22.3 (C-30), 19.6 (C-18), and 15.8 (C-19, 21-OCH_2_*CH*_3_). All the above NMR data suggested that compound **1** possess an apotirucallane-type triterpenoid skeleton [7,16,17]. Analysis of the HMBC spectrum (Figure 2) demonstrated the isovaleryl ester group was located in C-3 by the cross-signal from H-3 (*δ*_H_ 4.65) to C1′ (*δ*_C_ 174.6), H-4′ (*δ*_H_ 0.96) to C-3′ (*δ*_C_ 44.7) and C-2′ (*δ*_C_ 25.3), H-5′ (*δ*_H_ 0.96) to C-3′ (*δ*_C_ 44.7) and C-2′ (*δ*_C_ 25.3), and H-29 (*δ*_H_ 1.09) and H-28 (*δ*_H_ 0.86) to C-3 (*δ*_C_ 79.7). The HMBC correlation (Figure 2) between H-30 (*δ*_H_ 0.93) and C-7 (*δ*_C_ 73.7) indicated that the hydroxyl group was located at C-7 position. The HMBC correlations (Figure 2) between H-15 (*δ*_H_ 5.44) and C-16 (*δ*_C_ 35.9), C-17 (*δ*_C_ 59.3) revealed the presence of a double bond between C-14 and C-15. All the data above indicated that the basic structure of **1** was similar to those of dictamnin B [17], except that the presence of 21-OCH_3_ was replaced by 21-OCH_2_CH_3_ and 26-CH_2_OH was replaced by 26-CH_3_ in **1**. This was additionally confirmed by the key HMBC correlations (Figure 2) from H-21 (*δ*_H_ 4.92) to 21-O*CH*_2_CH_3_ (*δ*_C_ 64.8), 21-OCH_2_*CH*_3_ (*δ*_H_ 1.20) to 21-O*CH_2_*CH_3_ (*δ*_C_ 64.8). The HMBC correlations (Figure 2) between the methyl proton signal H-26 (*δ*_H_ 1.26) and H-27 (*δ*_H_ 1.20) correlated with C-24 (*δ*_C_ 78.4), C-25 (*δ*_C_ 73.9), which indicated that the two methyl groups should be located at C-26 and C-27 position, respectively. The NOESY correlations (Figure 2) between H-3/CH_3_-29, H-5/CH_3_-28, H-7/CH_3_-30, H-9/CH_3_-18, H-17/CH_3_-30, CH_3_-19/CH_3_-29, and CH_3_-19/CH_3_-30 showed that isovaleryl ester group, H-5, OH-7, H-9, CH_3_-18, and CH_3_-28, was *α*-oriented, whereas H-17, CH_3_-19, CH_3_-29, and CH_3_-30 were *β*-oriented. Furthermore, The NOESY correlations (Figure 2) were also observed between H-17/H-21, H-21/H-22*β*, H-22*α*/H-23, H-20/H-23, and H-23/H-24, which were possible only when **1** possessed 20*S*, 21*R*, 23*R*, and 24*R* configuration. Therefore, compound **1** was identified as toonasinensin A.

Compound **2** was obtained as a white amorphous powder. Its molecular formula was determined as C_37_H_60_O_7_ by the positive HR-ESI-MS (Appendix A) at *m*/*z* 615.42554 [M–H]^−^ (calculated. 615.42663). The ^1^H-NMR spectrum (Appendix A) displayed an olefinic proton at *δ*_H_ 5.77 (1H, s, H-2′); five hydroxy methine proton at *δ*_H_ 4.96 (1H, m, H-21), 4.62 (1H, m, H-3), 4.24 (1H, m, H-23), 3.72 (1H, br s, H-7), and 3.22 (1H, m, H-24); one hydroxy methylene proton at *δ*_H_ 3.68 (1H, m, 21-O*CH*_2_CH_3_a), 3.40 (1H, m, 21-O*CH*_2_CH_3_b); and nine methyl groups at *δ*_H_ 2.17 (3H, s, H-5′), 1.92 (3H, s, H-4′), 1.25 (3H, s, H-27), 1.20 (3H, s, H-26), 1.05 (3H, s, H-30), 0.95 (3H, overlap, H-19), 0.92 (3H, overlap, 21-OCH_2_*CH*_3_), 0.91 (3H, overlap, H-29), and 0.83 (3H, s, H-28). The ^13^C-NMR spectrum (Appendix A) showed one carbonyl group at *δ*_C_ 166.7 (C-1′); one olefinic carbon group at *δ*_C_ 156.2 (C-3′) and 116.2 (C-2′); seven oxygenated carbons at *δ*_C_ 109.1 (C-21), 77.4 (C-3), 77.1 (C-24), 75.9 (C-23), 74.0 (C-7), 72.5 (C-25), and 67.7 (21-O*CH*_2_CH_3_); and nine methyl groups at *δ*_C_ 26.8 (C-28), 26.0 (C-4′), 25.9 (C-27), 24.1 (C-26), 20.9 (C-29), 19.1 (C-5′), 18.8 (C-30), 14.7 (C-19), and 12.8 (21-OCH_2_*CH*_3_). All the data above indicated that the basic structure of **2** was also an apotirucallane-type triterpenoid, and the NMR data were similar to those of apotirucallane triterpenoid compound **1** from the literature [16], except that the presence of 21-OCH_3_ was replaced by 21-OCH_2_CH_3_ in **2**, which was confirmed by the key HMBC correlations (Figure 2) from H-21 (*δ*_H_ 4.96) to 21-O*CH*_2_CH_3_ (*δ*_C_ 67.7) and 21-OCH_2_*CH*_3_ (*δ*_C_ 12.8). The configuration of compound **2** was determined on the NOESY correlations (Figure 2) between H-3/CH_3_-29, H-5/CH_3_-28, H-7/CH_3_-30, H-17/CH_3_-30, H-17/H-21, H-21/H-22*β*, H-22*α*/H-23, and H-23/H-24. Consequently, the structure of compound **2** was named as toonasinensin B.

The other known triterpenoids were identified as toonasinensin C (**3**) [7], toonasinensin D (**4**) [16], toonasinensin E (**5**) [18,19], 21β-*O*-methylmelianodiol (**6**) [20,21], 21α-*O*-methylmelianodiol (**7**) [22], cycloeucalenol (**8**) [23], and 24-methylenecycloartanol (**9**) [24,25].

All compounds (**1**–**9**) were assessed for their cytotoxicity against GMCs. Compounds **2**, **4**, **6**, **7**, and **9** showed no cytotoxicity against GMCs at 80 μM compared with the normal group (NG) (Appendix A). The proliferation of GMCs is the main pathologic feature of DN, and a variety of stimuli were found to be associated with that proliferation, including high-glucose [26,27,28,29]. We investigated the effects of compounds (**2**, **4**, **6**, **7**, and **9**) on GMCs proliferation. As shown in Figure 3, GMCs proliferation became significantly greater than in the NG upon exposure to the high-glucose group (HG). However, treatment with compounds (**2**, **4**, **6**, **7**, and **9**) significantly inhibited high-glucose induced GMCs proliferation.

Oxidative stress caused by high glucose is a major process in the progression of DN. In order to determine the effect of high glucose on oxidative stress of GMCs, we determined the levels of superoxide dismutase (SOD), malondialdehyde (MDA), and ROS. The GMCs were treated at concentrations of 10, 30, and 50 μM. The results showed that compounds **2**, **6**, and **7** could render the level of SOD higher than in the HG (Figure 4). Compounds **2**, **6**, and **7** could render the levels of MDA and ROS lower than in the HG (Figure 5 and Figure 6). These results indicated that they could significantly reduce oxidative stress of GMCs. A preliminary structure-activity relationship indicated that apotirucallane-type triterpenoids (**2**, **6**, and **7**) showed significant antioxidant effects with respect to DN; nevertheless cycloartane-type triterpenoids (**8**–**9**) had no antioxidant activities. Even more interesting was that compounds **6** and **7** possessed similar activities, perhaps because the stereochemistry of C-21 could not affect the strength of antioxidant activities. In this way, apotirucallane-type triterpenoids have the potential for further development and research.

## 3. Materials and Methods

### 3.1. General Experimental Procedures

High-resolution electrospray ionization mass spectra (HR-ESI-MS) were obtained using a Bruker microsoft time-of-flight QII mass spectrometer (Bruker Daltonics, Fremont, CA, USA). The NMR spectra were recorded using Bruker AV 500MHz spectrometer (Bruker, Fällanden, Switzerland). Optical rotation was measured using a Rudolph Autopol I automatic polarimeter (Rudolph Research Analytical, Hackettstown, NJ, USA). Column chromatography was performed using silica gel (200–300 mesh, Branch of Qingdao Haiyang Chemical Co., Ltd., Qingdao, China) and Sephadex LH-20 (Shanghai Yuanye Biological Technology Co., Ltd., Shanghai, China). Lichroprep RP-18 gel (40–60 μm) was purchased from Merck KGaA (Darmstadt, Germany). Thin layer chromatography (TLC) was performed with precoated silica gel GF 254 glass plates (100 × 200 mm, Branch of Qingdao Haiyang Chemical Co., Ltd.). All other chemicals and solvents were analytical grade and used without further purification.

### 3.2. Plant Material

The pericarp of *T. sinensis* was collected by the Jinan Shengke Technology Company of China and identified by Prof. Chongmei Xu. A voucher specimen (voucher number: WF-YXY-1712) has been deposited at the Pharmacognosy Laboratory of the School of Pharmacy, Weifang Medical University.

### 3.3. Extraction and Isolation

Dried pericarp of *T. sinensis* (20 kg) was extracted three times with 95% EtOH (100 L × 3 times) and heated for 10 h. The combined extracts were concentrated under a vacuum to obtain a crude extract (854 g). The crude extract was suspended in H_2_O (3 L) and partitioned sequentially with CH_2_Cl_2_, EtOAc, and *n*-BuOH (3 L × 3 times in each case). The CH_2_Cl_2_ extract (147 g) was rested after evaporations. The CH_2_Cl_2_ extract was dissolved in the mixed solvent of CH_2_Cl_2_-MeOH, and silica gel (185 g) was added to conduct dry sample mixing before further fractionation. The CH_2_Cl_2_ extract (147 g) was purified by silica gel column chromatography and eluted with a gradient of petroleum ether:EtOAc (30:1, 10:1, 5:1, 2:1, *v*/*v*) and CH_2_Cl_2_:MeOH (20:1, 5:1, *v*/*v*) to generate 6 fractions (Fr. A-F). Fractions that flowed out of the column chromatography were combined based on their TLC patterns in the whole separation experiment. Fr. A (20 g) was separated chromatographically using octade-cylsilyl (ODS) and eluted with MeOH-H_2_O (from 40% to 100%) to produce 12 subfractions (Fr. A1-A12). Fr. A7 (4.6 g) was separated using a gradient of petroleum ether:EtOAc (from 80:1 to 40:1) to produce 5 fractions (Fr. A71-A75). Fr. A74 (1.4 g) was separated using Sephadex LH-20 to give compound **3** (54.5 mg). Fr. B (30 g) was separated chromatographically using ODS and eluted with MeOH-H_2_O (from 40% to 100%) to produce 9 subfractions (Fr. B1-B9). Fr. B8 (9.1 g) was separated using silica gel with a gradient of petroleum ether:EtOAc (from 45:1 to 30:1) to produce 4 fractions (Fr. B81-B84). Fr. B83 was separated using silica gel with a gradient of petroleum ether:EtOAc (from 80:1 to 60:1) to give compound **9** (21.3 mg). Fr. B84 was separated using silica gel with a gradient of petroleum ether:EtOAc (from 70:1 to 50:1) to produce 10 fractions (Fr. B841-B8410). Fr. B848 was separated using ODS and eluted with MeOH-H_2_O (from 88% to 100%) to give compound **8** (142.2 mg). Fr. C (20 g) was separated chromatographically using ODS and eluted with MeOH-H_2_O (from 30% to 100%) to produce 10 subfractions (Fr. C1-C10). Fr. C9 (7.2 g) was separated using silica gel with a gradient of petroleum ether:EtOAc (from 10:1 to 5:1) to produce 4 fractions (Fr. C91-C94). Fr. C92 (1.2 g) was separated using silica gel with a gradient of petroleum ether:EtOAc (from 9:1 to 7:1) to give compound **5** (17.9 mg). Fr. C93 (1.2 g) was separated using silica gel with a gradient of petroleum ether:EtOAc (from 9:1 to 7:1) to produce 4 fractions (Fr. C931-C934). Fr. C931 (0.4 g) was separated using ODS and eluted with MeOH-H_2_O (from 75% to 85%) to give compound **6** (17.2 mg). Fr. C932 (0.35 g) was separated using ODS and eluted with MeOH-H_2_O (from 75% to 80%) to give compound **7** (12.9 mg). Fr. D (20 g) was separated using ODS and eluted with MeOH-H_2_O (from 30% to 90%) to produce 13 subfractions (Fr. D1-D13). Fr. D9 (5.4 g) was separated using silica gel with a gradient of petroleum ether:EtOAc (from 7:1 to 4.5:1) to produce 5 fractions (Fr. D91-D95). Fr. D94 (2.6 g) was isolated using Sephadex LH-20 to obtain compound **1** (12 mg). Fr. D10 (8 g) was separated using silica gel with a gradient of petroleum ether:EtOAc (from 6:1 to 3:1) to produce 5 fractions (Fr. D101-D105). Fr. D101 (1.5 g) was isolated using Sephadex LH-20 to obtain compound **4** (17.9 mg). Fr. D105 was isolated using ODS and eluted with MeOH-H_2_O (75%) to give compound **2** (6 mg) (Appendix A).

#### 3.3.1. Toonasinensin A (1)

White amorphous powder; C_37_H_62_O_7_;
${\left[\alpha \right]}_{D}^{24}$
–20.64 (*c* 0.17, MeOH); HR-ESI-MS *m*/*z* 617.44177 [M –H]^−^ (calcd. 617.44228); ^1^H-NMR (methanol-*d*_4_, 500 MHz) and ^13^C-NMR data (methanol-*d*_4_, 125 MHz), which were unambiguously assigned by distortionless enhancement by polarization transfer (DEPT) 135, HMQC, HMBC, and NOESY experiments (Appendix A) (see Table 1).

#### 3.3.2. Toonasinensin B (2)

White amorphous powder; C_37_H_60_O_7_;
${\left[\alpha \right]}_{D}^{24}$
–14.73 (*c* 0.08, MeOH); HR-ESI-MS *m*/*z* 615.42554 [M –H]^−^ (calcd. 615.42663); ^1^H-NMR (methanol-*d*_4_, 500 MHz) and ^13^C-NMR data (methanol-*d*_4_, 125 MHz), which were unambiguously assigned by DEPT 135, HMQC, HMBC, and NOESY experiments (Appendix A) (see Table 1).

### 3.4. Cytotoxicity Assay

Cytotoxic activity against GMCs (Keygen Biotechnology, Nanjing, China) was measured using the 3-(4, 5-dimethylthiazol-2-yl)-2, 5-diphenyl tetrazolium (MTT) method. GMCs (5 × 10^3^ cells/well) in Dulbecco′s Modified Eagle Medium (DMEM) with 10% fetal bovine serum (FBS) were plated into 96-well plates. Then GMCs were cultured with 5.6 mM glucose (normal group, NG) or NG medium in the presence of compounds (**1**–**9**) (80 μM) for 48 h, followed by the addition of 100 μL of the MTT solution (0.5 mg/mL) to each well and further incubation for 4 h. The medium was removed and the dark blue crystals in each well were dissolved in 100 μL dimethyl sulfoxide (DMSO). The absorbance of the wells was measured with a microplate reader at test and reference wavelength of 490 nm.

### 3.5. Cell Proliferation Assay

GMCs were plated into 96-well plates. Then GMCs were divided into NG, NG medium in the presence of mannitol (25 mM), 25 mM high glucose (high-glucose group, HG), HG medium in the presence of epalrestat (10 μM), and HG medium in the presence of compounds (**2**, **4**, **6**, **7**, and **9**) (5, 10, 20, 40, and 80 μM). Cell proliferation was measured using the MTT assay. Mannitol was used as osmotic pressure group and epalrestat was used as positive control.

### 3.6. In Vitro Antioxidant Activity of SOD, MDA, and ROS

#### 3.6.1. SOD

Superoxide dismutase is an important antioxidant enzyme defense in all organisms [30]. SOD levels were detected by Nanjing Jiancheng Bioengineering Institute assay kit. GMCs were cultured in a 6-well plate at 3 × 10^5^ cells/well and exposed to the compounds (10, 30, and 50 µM) for 48 h. Absorbance was recorded at 450 nm using microplate reader. SOD activity is expressed as (U/mL), where each unit represents the amount of enzyme. The disproportionation of 50% superoxide radicals needs to be revealed.

#### 3.6.2. MDA

MDA is an indicator of lipid peroxidation. MDA was detected by Nanjing Jiancheng Bioengineering Institute assay kit [31,32]. GMCs were cultured in a 6-well plate at 3 × 10^5^ cells/mL and exposed to the compounds (10, 30, and 50 µM) for 48 h. Absorbance was recorded at 532 nm using microplate reader.

#### 3.6.3. ROS

Medium and high concentrations of ROS induce cell apoptosis and even necrosis through cell oxidative stress reaction [33,34]. GMCs were cultured in a 96-well plate at 5 × 10^3^ cells/well and exposed to the compounds (10, 30, and 50 µM) for 48 h. The cells’ supernatant was discarded and determined ROS by reactive oxygen species assay kit (Beijing Solarbio Science and Technology Co., Ltd., Beijing, China). Measured fluorescence intensity with fluorescence microplate reader at excitation and emission wavelengths of 488 and 525 nm.

### 3.7. Statistical Analysis

Statistical differences between two groups were analyzed by the T-test and differences between multiple groups of data were analyzed by one-way ANOVA with the prism software (GraphPad, San Diego, CA), and the data were expressed as the mean ± SD of three independent experiments. A *p* value of less than 0.05 was considered statistically significant.

## 4. Conclusions

Nine compounds were isolated from the pericarp of *T*. *sinensis*, including two previously unreported apotirucallane-type triterpenoids, toonasinensin A (**1**) and toonasinensin B (**2**); five known apotirucallane-type triterpenoids (**3**–**7**); and two known cycloartane-type triterpenoids (**8**–**9**). Compounds **2**, **4**, **6**, **7**, and **9** were found to significantly inhibit high-glucose induced GMCs proliferation. Compounds **2**, **6**, and **7** were able to significantly increase the vitality of SOD and reduce the levels of MDA and ROS. In summary, compounds **2**, **6**, and **7** were able to prevent DN by reducing oxidative stress, indicating that *T*. *sinensis* is worthy of further exploration to find more novel constituents with potential bioactivity.

## Figures and Tables

**Figure 1 molecules-25-00801-f001:**
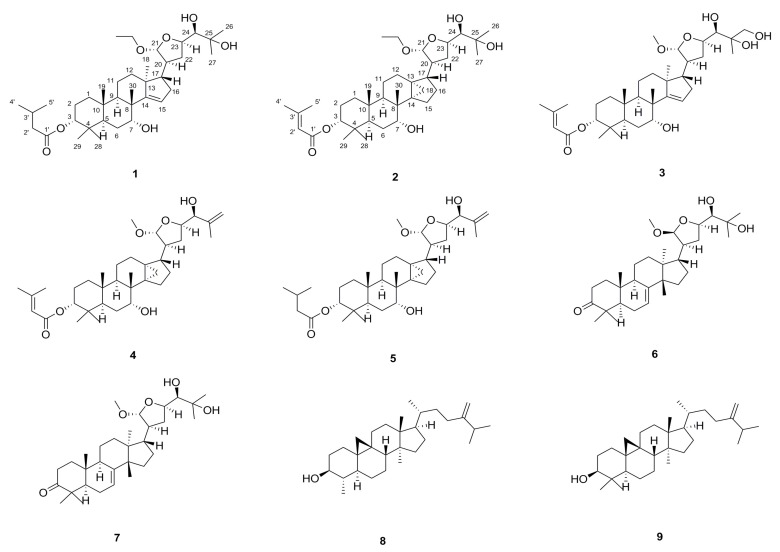
Structures of compounds **1**–**9** from the pericarp of *T. sinensis*.

**Figure 2 molecules-25-00801-f002:**
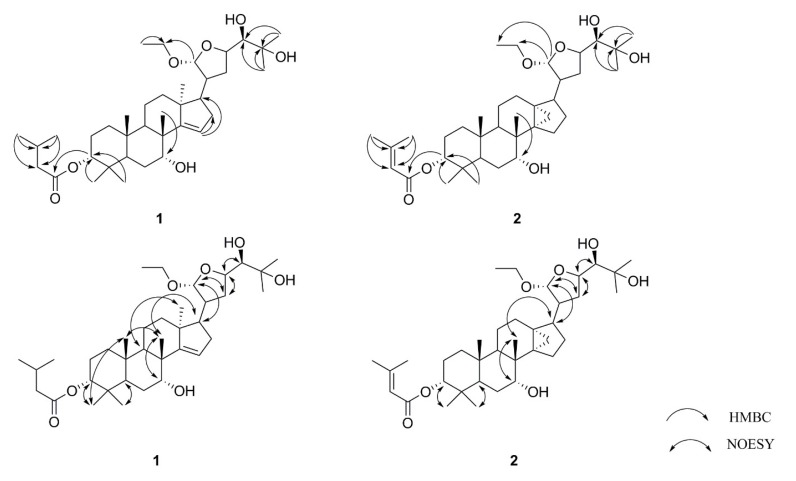
Key HMBC and NOESY correlations of compounds **1**–**2**.

**Figure 3 molecules-25-00801-f003:**
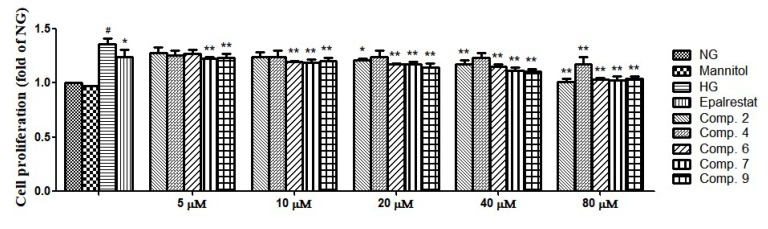
**The** rat glomerular mesangial cells (GMCs) were pretreated with various concentrations of compounds **2**, **4**, **6**, **7**, and **9** (5, 10, 20, 40, and 80 μM) and incubated for 48 h. GMCs proliferation was determined using the 3-(4, 5-dimethyl-2-thiazolyl)-2, 5-diphenyl-2-H-tetrazolium bromide (MTT) assay. Values are expressed as mean ± SD of three independent experiments, with ^#^
*p* < 0.01 relative to the 5.6 mM glucose (normal group, NG), and ** *p* < 0.01, * *p* < 0.05 relative to the 25 mM high glucose (high-glucose group, HG).

**Figure 4 molecules-25-00801-f004:**
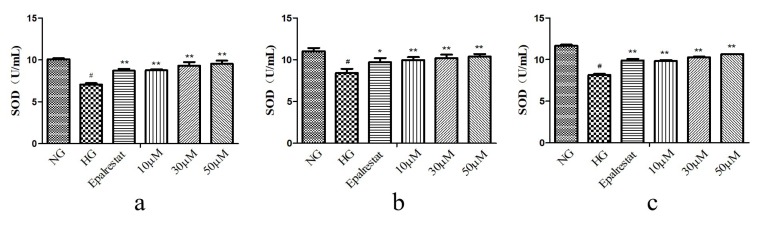
Compounds **2** (**a**), **6** (**b**), and **7** (**c**) increased the levels of superoxide dismutase (SOD) induced by high glucose in GMCs. GMCs were incubated with or without compounds **2**, **6**, and **7** (10, 30, and 50 µM) in the 5.6 mM glucose (normal group, NG) or 25 mM high glucose (high-glucose group, HG) for 48 h. ^#^
*p* < 0.01 compared with the NG, and ** *p* < 0.01, * *p* < 0.05 relative to the HG.

**Figure 5 molecules-25-00801-f005:**
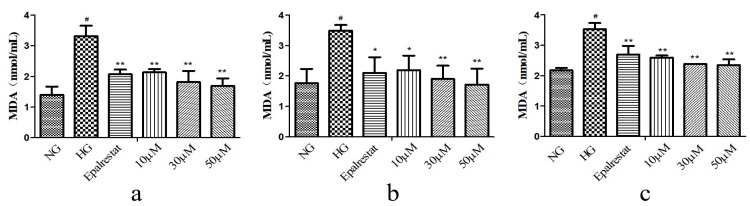
Compounds **2** (**a**), **6** (**b**), and **7** (**c**) inhibited the marker of oxidative stress malondialdehyde (MDA) by high- glucose in GMCs. GMCs were incubated with or without compounds **2**, **6**, and **7** (10, 30, and 50 µM) in the 5.6 mM glucose (normal group, NG) or 25 mM high glucose (high-glucose group, HG) for 48 h. ^#^
*p* < 0.01 relative to the NG, and ** *p* < 0.01, * *p* < 0.05 compared with the HG.

**Figure 6 molecules-25-00801-f006:**
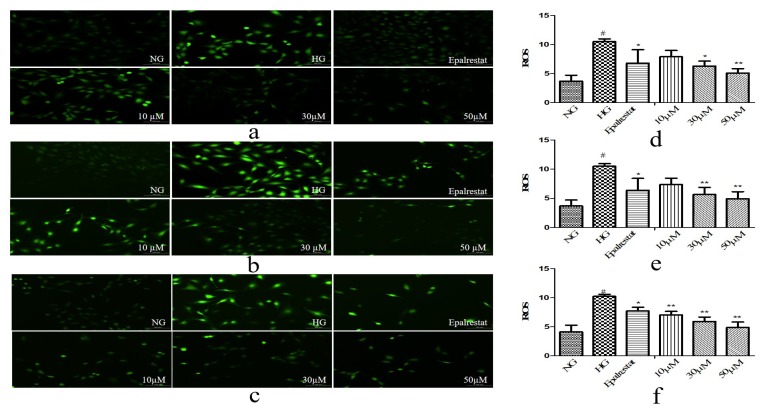
Compounds **2** (**a**,**d**), **6** (**b**,**e**), and **7** (**c**,**f**) inhibited the marker of oxidative stress reactive oxygen species (ROS) induced by high glucose in GMCs. GMCs were incubated with or without compounds **2**, **6**, and **7** (10, 30, and 50 µM) under the 5.6 mM glucose (normal group, NG) or 25 mM high glucose (high-glucose group, HG) for 48 h. ^#^
*p* < 0.01 relative to the NG, and ** *p* < 0.01, * *p* < 0.05 relative to the HG.

**Table 1 molecules-25-00801-t001:** The ^1^H- and ^13^C-NMR data (500 and 125 MHz) of compounds **1** and **2** (*δ* in ppm) in methanol-*d*_4._

Pos.	1		2	
	*δ* _C_	*δ*_H_ (*J* in Hz)	*δ* _C_	*δ*_H_ (*J* in Hz)
1	34.6	1.40 m, 1.27 m	33.6	1.38 m, 1.21 m
2	23.8	1.95 m, 1.63 m	22.5	1.97 m, 1.54 m
3	79.7	4.65 m	77.4	4.62 m
4	37.3	-	36.3	-
5	43.0	2.07 m	41.2	2.01 m
6	23.8	2.03 m, 1.54 m	24.5	1.68 m, 1.57 m
7	73.7	3.94 br s	74.0	3.72 br s
8	45.2	-	39.1	-
9	43.2	2.06 m	44.1	1.39 m
10	38.8	-	37.2	-
11	17.5	1.69 m, 1.54 m	16.1	1.32 m
12	34.4	1.87 m, 1.49 m	25.4	1.87 m
13	48.1	-	28.4	-
14	162.7	-	36.1	-
15	120.3	5.44 br s	25.6	1.92 m, 1.55 m
16	35.9	2.16 m	25.7	1.67 m
17	59.3	1.71 m	48.6	1.99 m
18	19.6	1.11 s	13.5	0.72 m, 0.51 m
19	15.8	0.98 s	14.7	0.95 overlap
20	47.5	2.33 m	49.1	2.06 m
21	109.7	4.92 d (3.3)	109.1	4.96 m
22	36.1	1.94 m, 1.65 m	31.7	1.59 m, 1.51 m
23	77.1	4.27 m	75.9	4.24 m
24	78.4	3.24 m	77.1	3.22 m
25	73.9	-	72.5	-
26	27.0	1.26 s	24.1	1.20 s
27	25.3	1.20 s	25.9	1.25 s
28	27.5	0.86 s	26.8	0.83 s
29	28.6	1.09 s	20.9	0.91 overlap
30	22.3	0.93 s	18.8	1.05 s
1′	174.6	-	166.7	-
2′	25.3	1.19 m	116.2	5.77 s
3′	44.7	2.23 m	156.2	-
4′	22.8	0.96 s	26.0	1.92 s
5′	22.8	0.96 s	19.1	2.17 s
21-O*CH*_2_CH_3_	64.8	3.76 m, 3.46 m	67.7	3.68 m, 3.40 m
21-OCH_2_*CH*_3_	15.8	1.20 s	12.8	0.92 overlap

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
