# Peer review of "Two New Apotirucallane-Type Triterpenoids from the Pericarp of Toona sinensis and Their Ability to Reduce Oxidative Stress in Rat Glomerular Mesangial Cells Cultured under High-Glucose Conditions"

_molecules, 2020, doi:10.3390/molecules25040801_

Round 1

Reviewer 1 Report

Authors have isolated and elucidated the structure of nine triterpenoids from Toona sinensis pericarp, from two distinct types, and evaluated their antioxidant effects with respect to diabetes nephropathy. Nevertheless, only two of them are new. Some issues need attention.

Discussion of the results from the bioactivity assays is inexistent. Authors should try to establish structure/activity relationships.

Why haven’t the authors tested the same compounds’ concentrations in both the cytotoxicity and the antioxidant assays?

One cannot understand if the two new compounds are being described for the first time in this pericarp, in the species or in the Genus.

The origin of the glomerular mesangial cells is not given. In addition, item 3.5 is a repetition of 3.4.

Item 3.7. Which was the test used for statistical analysis and how was it chosen? Information about normality of distribution and homogeneity of variance of the data should be provided.

I find issues regarding English that should be addressed with the help of a native speaker or English language editing service.

Author Response

Dear Editors and Reviewers:

Thank you for your letter and for the reviewers’ comments concerning our manuscript entitled “Two new apotirucallane-type triterpenoids from the pericarp of Toona sinesis and their ability to reduce oxidative stress in rat glomerular mesangial cells cultured under high-glucose conditions” (ID: molecules-713445).

Those comments are valuable and very helpful for revising and improving our paper, as well as the guiding significance to our researches. We have studied comments carefully and have made correction which we hope meet with approval. Revised portions are marked in colours in the paper.

Response to Reviewer 1 Comments

Point 1: Discussion of the results from the bioactivity assays is inexistent. Authors should try to establish structure/activity relationships.

Response 1: Thank you for your careful review, according to the reviewer's comment, a preliminary structure-activity relationship was established, which indicated that apotirucallane-type triterpenoids (2, 6, and 7) showed significant antioxidant effects with respect to DN, nevertheless cycloartane-type triterpenoids (89) had no antioxidant activities. Even more interesting was that compounds 6 and 7 possessed similar activities, perhaps because the stereochemistry of C-21 could not affect the strength of antioxidant activities, which had been added in the manuscript.

Point 2: Why haven’t the authors tested the same compounds’ concentrations in both the cytotoxicity and the antioxidant assays?

Response 2: Usually, before to measure the antioxidant activities, we would inspect the cytotoxicity of chemical compounds in glomerular mesangial cells, in order that ensure the accuracy of biological results. Therefore, we will choose the maximum dosing concentration of cell proliferation assay as the dosing concentration of cytotoxic measurement. In this study, we choose five concentrations (5, 10, 20, 40, 80 μM) for the cell proliferation assay, hence, 80 μM was set as dosing concentration of cytotoxicity assay. On the basis of the cell proliferation assay, we further investigated the antioxidant activities of the compounds with respect to DN to demonstrate the effects on the levels of oxidative stress factors SOD, MDA and ROS, therefore, we chose high (50 μM), medium (30 μM), and low (10 μM) three concentrations as dosing concentrations of antioxidant activities of SOD, MDA, and ROS according to the concentrations of the cell proliferation assay.

Point 3: One cannot understand if the two new compounds are being described for the first time in this pericarp, in the species or in the Genus.

Response 3: The two new compounds are unreported in previous studies.

Point 4: The origin of the glomerular mesangial cells is not given. In addition, item 3.5 is a repetition of 3.4.

Response 4: The rat glomerular mesangial cells (GMCs) were purchased from Keygen Biotechnology (Nanjing, China), which had been added in the manuscript. We are very sorry for confuse the reviewer about item 3.4 and 3.5. Although the MTT assay was used in item 3.4 and 3.5, and the groups of them were different, so we corrected them in the manuscript.

The statements of “Then GMCs were cultured with compounds (80 μM) for 48 h” was corrected as “Then GMCs were cultured with 5.6 mM glucose (normal group, NG) or NG medium in the presence of compounds (19) (80 μM) for 48 h”.,

The item 3.5 was corrected as “GMCs were plated into 96-well plates. Then GMCs were divided into NG, NG medium in the presence of Mannitol (25 mM), 25 mM high glucose (high-glucose group, HG), HG medium in the presence of Epalrestat (10 μM), and HG medium in the presence of compounds (2, 4, 6, 7, and 9) (5, 10, 20, 40, and 80 μM). Cell proliferation was measured using the MTT assay. Mannitol was used as osmotic pressure group and epalrestat was used as positive control.”

Point 5: Item 3.7. Which was the test used for statistical analysis and how was it chosen? Information about normality of distribution and homogeneity of variance of the data should be provided.

Response 5: Statistical differences between two groups were analyzed by the T-test and differences between multiple groups of data were analyzed by one-way ANOVA with the prism software (GraphPad, San Diego, CA), and the data were expressed as the mean ± SD of three independent experiments. A P value of less than 0.05 was considered statistically significant.

Point 6: I find issues regarding English that should be addressed with the help of a native speaker or English language editing service.

Response 6: Thank you for the careful review, according to the reviewer's comment, the English language had been corrected with LetPub in this manuscript.

We tried our best to make some modifications in the manuscript.  These changes will not influence the content and framework of the paper. We appreciate for Editors/Reviewers’ warm work, and hope that the correction will meet with approval. Once again, thank you very much for your comments and suggestions.

Sincerely yours,

Wanzhong Li

February 6, 2020

Reviewer 2 Report

Authors of the publication „Two new apotirucallane-type triterpenoids from the pericarp of Toona sinesis and their ability to reduce oxidative stress in rat glomerular mesangial cells cultured under high-glucose conditions” isolated two previously unknown triterpenoids and evaluated their cytotoxicity and antioxidant activity.

The search for new compounds that can be used in modern medicine is very important therefore those results deserve a lot of attention. In spite of that reviewer found few shortcomings in the manuscript which should be corrected before its publication.

Parameter which usually describes cytotoxicity is IC50 - not absorbance. Proliferative properties shouldn't be presented as an absorbance. All details related to the statistical analysis should be expanded. Which test was used to state statistical differences? What was the goal of using TLC in this study? Paragraph 3.3. should be better described. What volume of 95% EtOH was used to extract the raw material? Whether 147 g of extract CH2Cl2 was rest after evaporations or it was mass of concentrated extract? If extract CH2Cl2 has been evaporated to dryness, in what solvent it was dissolved before further fractionation? What amounts of stationary phases and what volumes of eluents were applied? I would like notice that without exhaustive description of this paragraph, reconstruction of extraction procedure is impossible. In my opinion for better understanding of the whole extraction and isolation procedure authors should make a graphical form that describes this process. Figures descriptions should include all used abbreviations (NG, HG) and information about statistical test which was used. For what purpose mannitol and epalrestat were used? Those informations absolutely should be in the text. References to figures presented in supplementary material should be included in the text. In my opinion authors should do HPLC analysis of the raw material (only few samples) to determine amounts of compounds which were isolated. It will allow to define real amounts of these compounds and efficiency of applied procedure. These researches are very important in context of practical possibilities of using these compounds in prevention diabetic nephropathy.

Author Response

    Dear Editors and Reviewers:

    Thank you for your letter and for the reviewers’ comments concerning our manuscript entitled “Two new apotirucallane-type triterpenoids from the pericarp of Toona sinesis and their ability to reduce oxidative stress in rat glomerular mesangial cells cultured under high-glucose conditions” (ID: molecules-713445).

    Those comments are valuable and very helpful for revising and improving our paper, as well as the guiding significance to our researches. We have studied comments carefully and have made correction which we hope meet with approval. Revised portions are marked in colours in the paper.

    We tried our best to make some modifications in the manuscript.  These changes will not influence the content and framework of the paper. We appreciate for Editors/Reviewers’ warm work, and hope that the correction will meet with approval. Once again, thank you very much for your comments and suggestions.

    Sincerely yours,

    Wanzhong Li

    February 6, 2020

Round 2

Reviewer 1 Report

Authors have generally addressed my previous comments and the work can be accepted for publication.

Author Response

Dear Reviewers:

Thank you for your letter and for the reviewers’ comments concerning our manuscript entitled “Two new apotirucallane-type triterpenoids from the pericarp of Toona sinesis and their ability to reduce oxidative stress in rat glomerular mesangial cells cultured under high-glucose conditions” (ID: molecules-713445).

We have studied comments carefully and have made correction which we hope meet with approval. Revised portions are marked in colours in the paper. We appreciate for your help, and thank you very much for your suggestions.

Sincerely yours,

Wanzhong Li

February 11, 2020

Reviewer 2 Report

The authors explained most of the doubts of the reviewer and introduced appropriate corrections. I recommend the manuscript to be published in Molecules.

However, minor corrections should be made:

What was the goal of using TLC in this study? All references to figures presented in supplementary material should be included in the text. Those
informations absolutely should be in the text.

Author Response

Dear Editors and Reviewers:

Thank you for your letter and for the reviewers’ comments concerning our manuscript entitled “Two new apotirucallane-type triterpenoids from the pericarp of Toona sinesis and their ability to reduce oxidative stress in rat glomerular mesangial cells cultured under high-glucose conditions” (ID: molecules-713445).

Those comments are valuable and very helpful for revising and improving our paper, as well as the guiding significance to our researches. We have studied comments carefully and have made correction which we hope meet with approval. Revised portions are marked in colours in the paper.

Response to Reviewer 2 Comments

Point 1: What was the goal of using TLC in this study?

Response 1: Fractions flowed out of the column chromatography were combined based on their TLC patterns in the whole separation experiment, which had been added in the manuscript.

Point 2: All references to figures presented in supplementary material should be included in the text.

Response 2: Thank you for your careful review, according to the reviewer′s comment, Figure S1-S16 presented in supplementary material had been included in the text.

We tried our best to make some modifications in the manuscript.  These changes will not influence the content and framework of the paper. We appreciate for Editors/Reviewers’ warm work, and hope that the correction will meet with approval. Once again, thank you very much for your comments and suggestions.

Sincerely yours,

Wanzhong Li

February 11, 2020